# Characterization and Control for the Laminar Flow of Liquid Polyurethane System in a Wide Angle Diffuser with Transversely Arrayed Obstacles

**Young Woo Son [1,2], Jong Hwi Lee [2] and Se-Myong Chang [2,\***]

1   College of Aerospace Enginnering, Nanjing University of Aeronautics and Astronautics, 29 Yudao St., Nanjing 210016, Jiangsu, China; syw@kunsan.ac.kr
2   Department of Mechanical Enginnering, Kunsan National University, 558 Daehak-ro, Gunsan 54150, Jeonbuk, Korea; vovbobvov@naver.com
\*   Correspondence: smchang@kunsan.ac.kr; Tel.: +82-63-469-4724

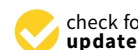

**Featured Application: Manufacturing Polyurethane Foams, Chemical Engineering.**

**Abstract:** In the manufacturing process of hard-board poly-urethane foams, the uniformity is a very important issue for the raw compound of the liquid poly-urethane system flow for the quality control of such products. One of the universal methods to generate more uniform flow is that some obstacles are located inside the diffuser at the end of injector. For the regime of non-Newtonian laminar flow, better flow uniformity can be achieved with the enhancement of mixing in the wake after the resistive obstacles. In this research, the parametric study is made for the gap interval between adjacent obstacle components as well as the cross-sectional shape with a computational fluid dynamics (CFD) technique. The flow fields around circular and elliptic cylinders are visualized for flow velocity and vorticity with the comparison of root-mean-square (RMS) error for the deviation of velocity at the outlet as a lumped parameter to estimate flow uniformity and mixing. When the blockage ratio is fixed 0.3 for the pipe of Reynolds number 58.5 based on its diameter, eliminating the effect of wall boundary ratio with the classical Blasius velocity profile, the RMS error is reduced 77% to 92% from the baseline case in the case of 60%-diameter gaps for the figure of circles and 2:1 longitudinal ellipse, respectively. The flow is visualized around obstacle components with vorticity as well as flow velocity where the three-dimensional components of vorticity vector are also elucidated in physics for the evolution of complex multi-dimensional flow wake.

**Keywords:** high-viscous fluid; flow uniformity; polyurethane board; mixing performance; non-Newtonian fluid

---

## 1. Introduction

Polyurethane foam is often used for adiabatic elements in the building construction owing to its excellent characteristics of such as mechanical strength, resistive endurance, elasticity, heat insulation, etc. Generally, a hard urethane board is manufactured through the four steps: mixing, reactions, injection, coagulation, and cutting. Briefly speaking on the process, first the precise pump transports the compound of liquids to the mixing chamber for the step of mixing and reaction. Then, the mixture is injected to be spread on the upper surface of a conveyer belt through an injection nozzle or diffuser before the reaction begins, which depends on the speed of both conveying and injection, also passed for the pressurization section to generate the foam by a series of counter-rotating rollers. Soon the rheological system is hardened for the coagulation at the atmospheric condition. The hard urethane board is produced as a commercial product after it is cut in a suitable size of length and width. In

the entire process, there are various parameters effecting on production and manufacture such as distribution of materials, size, and shape of bubbles inside the foam, related with the defect ratio as well as the overall mechanical property of the foam such as density, strength, hardness, etc. [1]. However, the most important key parameter, or the figure of merit is the uniformness of compound flow that results in quality of the product directly since it strongly induces all the above parameters. To achieve the uniform flow system, a transverse array of multiple obstacles is often located inside the diffuser of injector. Therefore, the main problem in this research is what will be the optimal configurations of these obstacle components in numbers, shapes, intervals, etc.

The mixture liquid for the raw material of polyurethane foams is that of high molecular weight matters, or a kind of non-Newtonian fluids with the characteristics of shear-thickening. Owing to their high viscosity levels, this material is generally regarded to be processed in the laminar flow conditions in the moderate flow speed. Various non-Newtonian fluids such as polymeric systems display viscoelastic behaviors, but so far only a few available literatures have been published either for the creeping flow past a single cylinder or over a periodic transverse array of cylinders across the main flow, which implies that the viscoelastic effects have not been well investigated as they are regarded as minor in the present flow configuration [2], but so far the researches on numerical analyses have been much published in the category of rheology using the index of power law [3–17]. Some of them manifested various physics related with flow such as drag coefficient, vortex shedding, heat transfer, etc., around single shape or many figures of circular or elliptic cylinders [18,19]. However, it is hard to find an investigation on the flow uniformity and the mixing characteristics through a nozzle or a diffuser containing multiple bodies of repetitive configurations. In the past, the authors studied some exercises similar problems of Newtonian flow, and an optimal configuration could be found for the flow uniformity in the mixing characteristics of laminar flow [20]. Therefore, this study will be the extension of them to a non-Newtonian version.

In this study, a commercial code for the numerical analysis, COMSOL Multiphysics 5.3a is used for the whole simulations for the injection diffuser flow model. As the flow regime of liquid polyurethane system lies in rheological pseudo-plasticity characteristics of a non-Newtonian fluid, a model of power law is applied for this numerical analysis where the flow viscosity is a power function of strain rate with empirical coefficients, which can be edited in the COMSOL source code.

## 2. Methods of Research

### 2.1. Model for Analysis

The geometrical configuration is shown for the present models in Figure 1 where the dimension data are presented in Table 1 [20]. The obstacles installed inside the diffuser are in shape of columns transversely blocking the cross-section where the diameter of circles or the short-axis length of ellipses is each fixed to 5 mm. The blockage ratio defined as the area ratio blocking the cross section at the transverse position of interest is set as a fixed value 0.3, and the multiple obstacles of circular or longitudinally elliptic cylinders are installed transversely inline, respectively, where the gap of obstacles are parameterized from 2 mm to 7 mm with 1 mm intervals (six cases each). Therefore, test cases are 12 in total for two kinds of obstacles: circular and 2:1 elliptic cylinders to analyze the flow uniformity and the mixing performance.

**Table 1.** Dimensions of diffuser elements.

| Symbols in Figure 1 | Value (Unit: mm; $\theta$ in Degrees) | Symbols in Figure 1 | Value (Unit: mm; $\theta$ in Degrees) |
|---|---|---|---|
| $D_i$ | 12 | $L_\theta$ | 70.7 |
| $L_t$ | 400 | $L_n$ | 94 |
| $w_1$ | 100 | $L_1$ | 30 |
| $w_2$ | 40 | $L_2$ | 64 |
| $\theta$ | 25° | $L_3$ | 24 |

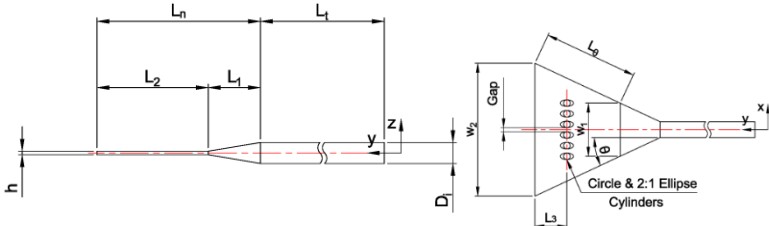

**Figure 1.** Analysis model of diffuser: side and planform view.

## 2.2. Governing Equations

Three-dimensional incompressible Navier–Stokes equations are used as the governing equations:

$$\frac{\partial \rho}{\partial t} + \nabla(\rho V) = 0 \tag{1}$$

$$\rho \left\{ \frac{\partial V}{\partial t} + (V \cdot \nabla)V \right\} = -\nabla p + \nabla(\mu \nabla V) \tag{2}$$

consisting of continuity equation, Equation (1), and momentum equations, Equation (2); $V$ is the velocity vector; $p$ is pressure; and $\rho$ is mean density of the compound.

The 'Laminar Fluid Flow' module in COMSOL Multiphysics is utilized to discretize Equations (1) and (2). The numerical method based on finite element method (FEM) used in this code adopts flexible generalized minimum residual (FGMRES) linearization and Petrov–Galerkin least-square artificial dissipation terms [21].

To analyze non-Newtonian flow, the rheological characteristics are reflected as a power-law model such as [22]:

$$\mu_{i,j} = m \left( \dot{\gamma}_{i,j} \right)^{n-1} \tag{3}$$

$$\dot{\gamma}_{i,j} = \left| \frac{\partial u_i}{\partial x_j} - \frac{\partial u_j}{\partial x_i} \right| \tag{4}$$

The dynamic viscosity $\mu$ in Equation (2) should be modified as the function of $\dot{\gamma}$ that is called shear rate like Equations (3) and (4) where the subscripts $i$ and $j$ denotes the directions in three dimension for the viscosity tensor in Equation (2), and m is the flow consistency index; the power n is the flow behavior index forced as constants for the majority of non-Newtonian fluid models.

## 2.3. Boundary Conditions

Boundary conditions are marked and pointed in Figure 2 where the obstacles are omitted in Figure 1, simply applied with no-slip conditions. The flow properties are listed in Table 2, and the mass flow rate is that used for the commercial 115 mm thickness urethane boards. The Reynolds number based on the inner diameter of the feeding pipe and its critical value are defined from the power law of Equation (3) as follows, respectively [23]:

$$Re_{d,PL} = 2^{3-n} \left( \frac{n}{3n+1} \right)^n \frac{V_{avg}^{2-n} D_i^n \rho}{m} \tag{5}$$

$$Re_{d,PL,critical} = 2100 \frac{(4n+2)(5n+3)}{3(3n+1)^2} \tag{6}$$

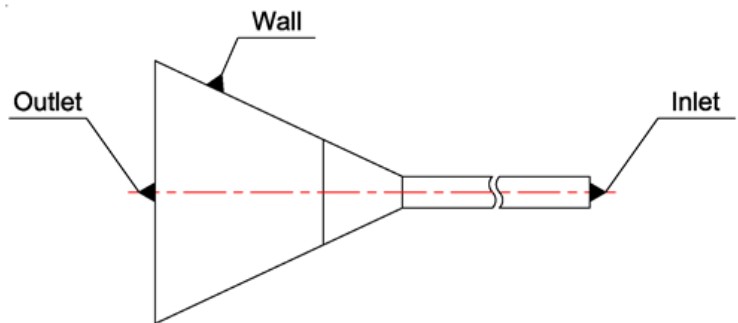

**Figure 2.** Schematic diagram of boundary condition.

**Table 2.** Boundary conditions.

| Properties | | Value | Unit |
|---|---|---|---|
| Mass flow rate at the inlet | | 779.5 | g/s |
| Gauge pressure at the outlet | | 0 | Pa |
| Density | | 1154 | kg/m$^3$ |
| Non-Newtonian | $m$ | 8.8562 | Pa·s$^n$ |
| viscosity | $n$ | 0.7669 | - |
| Properties [24] Wall velocity | | 0 (no slip) | - |

Recall that the critical Reynolds number becomes the same as that of a simple Newtonian pipe flow when we substitute $n = 1$ to Equation (6), which allow $\mu = m$ in Equation (5). For the present cases, the Reynolds number calculated from Equation (5) is 58.5, which is far less than 2225 from Equation (6), a calibrated critical Reynolds number from a generally accepted value in the internal pipe flow, so all the flow regime can be assumed as laminar in the present simulations.

*2.4. Mesh Generation*

Tetrahedral meshes are used in overall for all cases except for the near wall domains where the prism meshes are applied for the accuracy of computation, and hexahedral configuration of five mesh laminates to capture the boundary layer near from the feeding tube. The grid sensitivity is checked for the meshes shown in Figure 3a,b and the result reports that the total meshes in the whole computational domain is about 0.33 million that is concentrated 90% on the injection diffuser part.

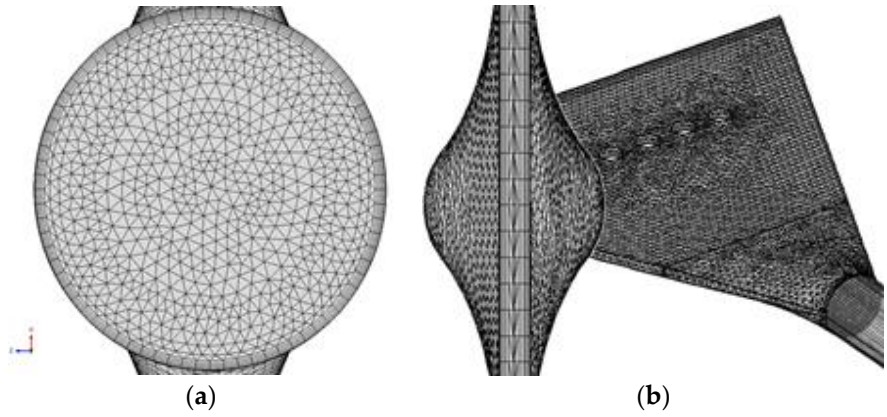

(**a**)            (**b**)

**Figure 3.** Grids system: (**a**) tube section, and (**b**) diffuser section.

To compare the Darcy friction coefficient ($f$) [25] for the various levels of grids, the Hagen–Poiseuille solution for a straight pipe is computed for the present meshes in Figure 3, and the average flow velocity ($V_{avg}$) is used for the computation of $f$:

$$f = -\frac{1}{4}\left(\frac{D_i}{L_t}\right)\left(\frac{2}{V_{avg}^2}\right)\frac{\Delta P}{\rho} \tag{7}$$

$$f = \frac{16}{Re_{d,PL}} \tag{8}$$

The pressure difference $\Delta P$ is measured for the known length of tube $L_t$, excluding the entrance length, and the Darcy frictional coefficient is calculated in Equation (7) to be compare with the analytic solution of Equation (8). The difference is checked within 1.6% error for the present level of grids, and, therefore, the inlet flow is at least regarded as satisfying the requirement of a proper scale mesh system.

## 3. Verification and Validation

### 3.1. Verification at the Tube Section

A simplified model is proposed to verify the present method for Hagen–Poiseuille solution in a non-Newtonian fluid: see Figure 4a. To get the fully developed flow domain, periodic flow condition is applied in both inlet and outlet boundaries, and the length $\Delta z$ is set 0.5 mm. The corresponding mesh system is plotted in Figure 4b, which is the identical level of the meshes in Section 2.4.

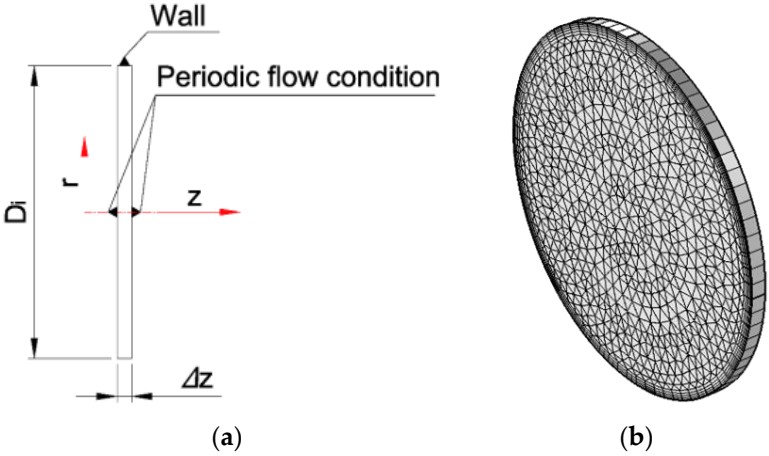

**Figure 4.** Verification at the tube section: (**a**) boundary condition, (**b**) grids system.

The classical solution of Hagen–Poiseuille [23,25] shows that the axisymmetric velocity distribution in a pipe should be given as a function of radial position, $r$:

$$u(r) = \frac{n}{n+1}\left(-\frac{dp}{dz}\frac{1}{2m}\right)^{\frac{1}{n}}\left(R^{\frac{n+1}{n}} - r^{\frac{n+1}{n}}\right) \tag{9}$$

where the pressure gradient $dp/dz < 0$ is specified, and the tube radius is $R$. From the comparison of the present numerical analysis with Equation (9) is shown in Figure 5, the maximum relative error is within 1.4%, and the present numerical method is verified for the tube-side computational region.

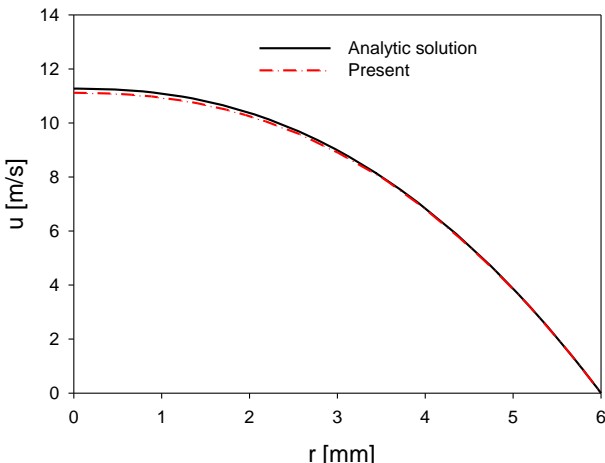

**Figure 5.** Comparison of Hagaen–Poiseuile flow and numerical result for non-Newtonian fluid.

### 3.2. Validation at the Diffuser wall Section

The expansion angle for the present diffuser is $25^{\circ}$, and we validated our numerical method for a far severer problem of a sudden expansion flow at a $90^{\circ}$ backward step, one of the experimental result from Denham and Patrick [26]. Although the fluid is Newtonian, the Reynolds number is calculated to 73 based on its two-dimensional step height, ranged between 10 and 100. In Figure 6, the relative error with the present result of streamwise velocity distribution, u, lies within 10% at all the wake stations from those of experimental result [26], so the computational method can be trusted as a proper one in the diffuser expansion part. The total meshes are 36,000, and uniform meshes are applied in $\Delta x = \Delta y = 0.05 \, [m]$. However, for the whole computational domain, there are located multiple cylindrical obstacles, this validation just works in the limited domain.

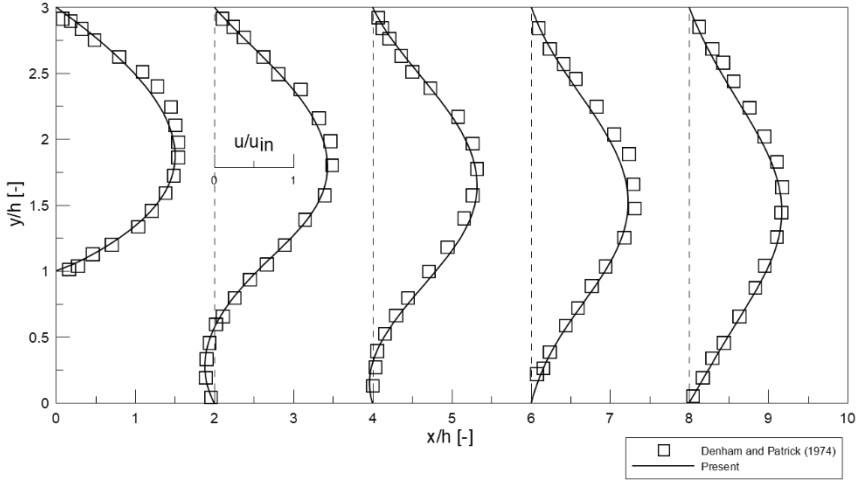

**Figure 6.** Non-dimension streamwise velocity profiles for $\mathrm{Re}_h \approx 73$.

### 3.3. Validation for the Body of Circular and Elliptic Shape

The flow around an element of obstacle is the key point of this research inside the diffuser. Computational domain and boundary conditions are such as Figure 7, setup to validate the flow around circular and elliptic obstacles [27], and Figure 8 shows the mesh used for numerical analysis where the mesh quality used for validation is equivalent to that used for the main problem.

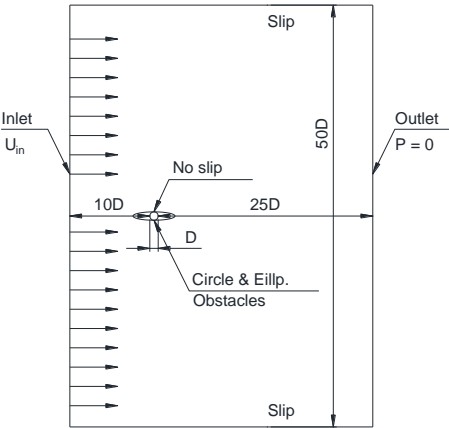

**Figure 7.** Computational domain and boundary conditions for the external flow around a body of circular or elliptic shape.

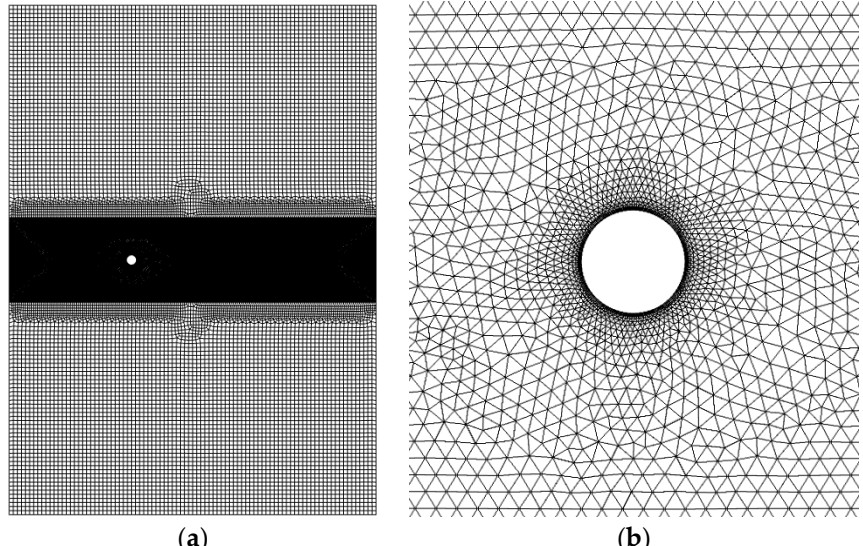

**Figure 8.** Grids system for the validation problem: (**a**) distribution of 31,904 elements, (**b**) zoomed view around a body.

The Reynolds number is 17, calculated from the definition as follow based on the short diameter of an obstacle where the flow around is that of symmetric separation for steady flow.

$$Re_d = \frac{\rho U_{in}^{2-n} D^n}{m} \tag{10}$$

where $U_{in}$ is the average inlet velocity.

The results are listed in Tables 3 and 4 for a circle and an ellipse, respectively. The $C_{DP}$ is the pressure drag coefficient, and $C_{DF}$ is the friction drag coefficient integrated from pressure and shear stress distribution at the surface of obstacles, respectively. Therefore, the total drag coefficient is the sum of these two drags, or $C_D = C_{DP} + C_{DF}$. The maximum error is estimated as 6%, so the present grid scale is regarded as proper.

**Table 3.** Numerical values of drag coefficients with power-law analysis around a circular obstacle.

| Coefficients | $C_{DP}$ | $C_{DF}$ | $C_D$ | $C_{DP}$ | $C_{DF}$ | $C_D$ | $C_{DP}$ | $C_{DF}$ | $C_D$ |
|---|---|---|---|---|---|---|---|---|---|
| | $n = 1$ | | | $n = 0.6$ | | | $n = 0.2$ | | |
| Present | 1.274 | 0.8362 | 2.111 | 1.355 | 0.6272 | 1.982 | 1.439 | 0.3372 | 1.776 |
| Sivakumar, P., et al. [16] | 1.201 | 0.8047 | 2.005 | 1.304 | 0.6137 | 1.918 | 1.484 | 0.3394 | 1.824 |
| Error (%) | 5.8 | 3.8 | 5.0 | 3.7 | 2.2 | 3.2 | 3.1 | 0.7 | 2.7 |

**Table 4.** Numerical values of drag coefficients with power-law analysis around an elliptic obstacle.

| Coefficients | $C_{DP}$ | $C_{DF}$ | $C_D$ | $C_{DP}$ | $C_{DF}$ | $C_D$ | $C_{DP}$ | $C_{DF}$ | $C_D$ |
|---|---|---|---|---|---|---|---|---|---|
| | $n = 1$ | | | $n = 0.6$ | | | $n = 0.2$ | | |
| Present | 0.9457 | 1.222 | 2.168 | 1.030 | 1.030 | 2.060 | 1.161 | 0.6817 | 1.843 |
| Sivakumar, P., et al. [16] | 0.8893 | 1.173 | 2.063 | 0.9900 | 1.013 | 2.003 | 1.184 | 0.7020 | 1.886 |
| Error (%) | 6.0 | 4.0 | 4.9 | 3.9 | 1.6 | 2.8 | 2.0 | 3.0 | 2.3 |

## 4. Results and Discussion

### 4.1. Flow Uniformity

The effect of obstacles is surveyed for its contribution on the flow uniformity. However, the edge flow accelerated from the blockage of the main central one effects on the unnecessary fluctuation near the side wall of diffuser. In Figure 9, the effect of boundary layers, or the shaded area neighboring to the side walls must be neglected to make a fair comparison.

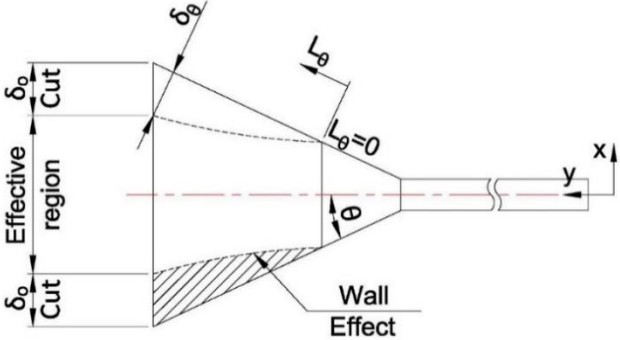

**Figure 9.** The schematic diagram for the cut regions near side walls.

The boundary layer on a flat plate is calculated with Blasius theory to simplify Equations (1) and (2) to the following incompressible boundary-layer equations in the two-dimension for the region of boundary layer near the side wall [20,22,24]:

$$\frac{\partial u}{\partial x} + \frac{\partial v}{\partial y} = 0 \tag{11}$$

$$u\frac{\partial u}{\partial x} + v\frac{\partial v}{\partial y} = -\frac{1}{\rho}\frac{dp}{dx} + v\frac{\partial}{\partial y}\left(\frac{\partial v}{\partial y}\right)^n \tag{12}$$

where the kinematic viscosity is simply defined as $v = \mu/\rho$ just like Newtonian fluid, which does not lose the generality even for such non-Newtonian fluids.

The stream function, $\psi$ is defined as, satisfying Equation (11),

$$u = \frac{\partial \psi}{\partial x}, v = -\frac{\partial \psi}{\partial y} \tag{13}$$

A dimensionless distance and the stream function are also defined as, just like those of Blasius,

$$\eta = y\left(\frac{U_\infty^{2-n}}{vx}\right)^{\frac{1}{n+1}} \tag{14}$$

$$\psi(\eta) = \left(vxU_\infty^{2n-1}\right)^{\frac{1}{n+1}} f(\eta) \tag{15}$$

Substituting Equation (15) into Equation (13), from a simple calculus, we get the velocity components such as

$$u = U_\infty f\prime \tag{16}$$

$$v = \frac{1}{n+1}\left(v\frac{U_\infty^{2n-1}}{x^n}\right)^{\frac{1}{n+1}} (\eta f' - f) \tag{17}$$

Note that there is no change in Equation (16) from the Newtonian fluid, and Equation (17) also falls down to Newtonian if $n = 1$.

As the outer flow from a flat plate, $dU_\infty/dx \approx 0$, and the pressure gradient term in Equation (12) is erased out. Equations (16) and (17) are substituted to Equation (12), the final ordinary equation is written down as

$$n(n+1)(f'')^{n-1}f''' + ff'' = 0 \tag{18}$$

The boundary condition of Equation (18) is also specified from the no-slip condition at the wall ($u = v = 0$, or $f\prime = f = 0$ at $\eta = 0$) and the inviscid condition at the outer edge of boundary layer ($u = U_\infty$, or $f\prime = 1$ at $\eta \to \infty$).

The solution of Equation (18) under the given boundary values are computed with 5th order Runge-Kutta method [22]: see the graphical result in Figure 10. This scheme works also in non-Newtonian fluid. Therefore, the thickness of boundary layer from the slanted side wall of diffuser in Figure 9 is easily found as the following scheme:

$$Re_{PL} = \frac{u_\theta^{2-n}L_\theta^n \rho}{m} \tag{19}$$

$$u_\theta = \frac{\dot{m}_i}{\rho L_\theta h}\cos\theta \tag{20}$$

$$\delta_\theta = L_\theta \frac{6.0}{Re_{PL}^{\frac{1}{n+1}}} \tag{21}$$

$$\delta_o = \frac{\delta_\theta}{\cos\theta} \tag{22}$$

where $\dot{m}_i$ is the mass flow rate. In Figure 11, the velocity profile at the diffuser outlet $f' = u/U_\infty = 0.99$ when $\eta \approx 6.0$, and Equation (21) is derived for the thickness of boundary layer. Therefore, the thickness in the transverse direction, or $\delta_O$ (see Figure 9) is computed approximately to 30 mm.

Thirteen cases including the baseline (with no obstacle) are defined from the parameter table (Table 5), and the gap size of multiple transverse obstacles located in Figure 1 is set from 2 to 7 mm with interval 1 mm, and the shapes are circle or 2:1 longitudinal ellipse where the diameter or the short axis length is set 5 mm.

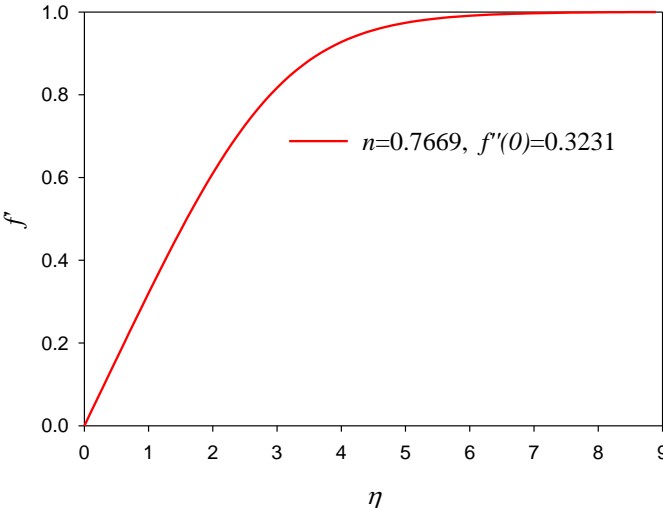

**Figure 10.** Velocity profiles for non-Newtonian fluid (*n* = 0.7669).

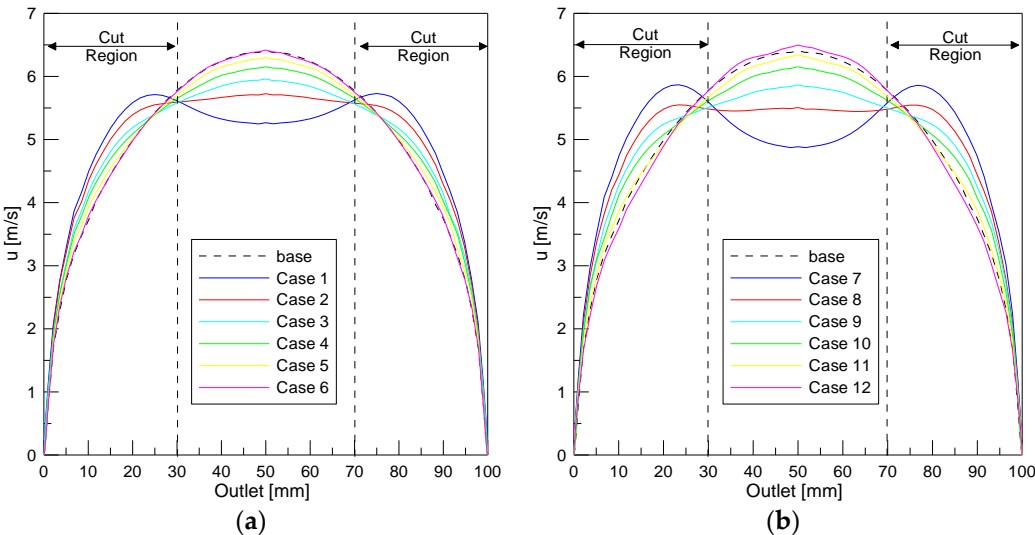

**Figure 11.** Velocity profile at the outlet with transverse obstacles: (**a**) circles, (**b**) 2:1 longitudinal ellipses.

**Table 5.** Computational cases.

| Case | Gap [mm] | Cross-Sectional Shape |
|---|---|---|
| 1~6 | 2~7 (interval: 1) | Circle |
| 7~12 | 2~7 (interval: 1) | 2:1 Ellipse (longitudinal) |

This thickness is reduced in Figure 11a,b for circle and ellipse, respectively, the outlet velocity profile, just filtering the velocity data to calculate the root-mean-square (RMS) of percentage deviation to determine the flow uniformity as follows:

$$\text{RMS} = \frac{u_{rms}}{u_{avg}} \times 100(\%) \tag{23}$$

$$u_{rms} = \sqrt{\frac{1}{n} \sum_{i=1}^{n} \left( u_i - u_{avg} \right)^2} \tag{24}$$

As seen in Figure 11a,b, Cases 1 and 7 reveals that there are overshoots of velocity at the boundary layer region, and the central velocity deficits due to the continuity of flow. In Figure 12, the difference of RMS value is not remarkable without cutting the edge or filter out the effect of the side diffuser walls, but with cutting the boundary data, Cases 2 and 8 are picked as the least RMS, which means the optimal flow uniformity, or the flattest central velocity profile at the wake of obstacles array.

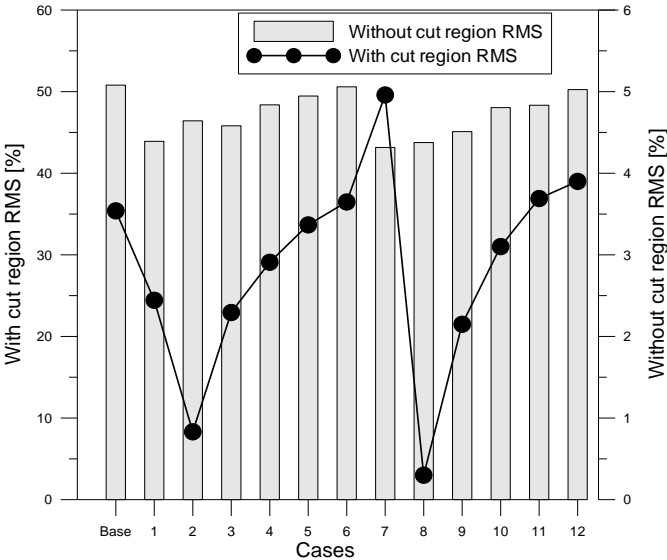

**Figure 12.** RMS error results by cases listed in Table 5.

### 4.2. Growth of Vorticity and Pressure Drop

Cases 2 and 8 in Figure 12, which are picked as the best cases for the flow uniformity, are compared for velocity and vorticity field in Figures 13 and 14. In Figure 13a,b, the flow is observed to be accelerated at the narrow gaps between each obstacle. The elliptic cylinders scatter the accelerated flow region far longer because its longitudinal axis distance is longer than that of circular ones. Figure 14a,b shows the vorticity, or the geometrical norm of curl $V$, to see the rotational mixed flow: for Case 2 (circular), it is observed that a microscopic flow separation develops a weak vortex shedding at the wake direction while the vortices generated from the boundary layer in Case 8 (elliptic) is shown to be attached at the long streamwise surface to delay the flow separation. It is understood that this slight difference results in more stable uniform flow at the wake of elliptic cylinders than that of circular ones. The penetrated vorticity providing the rotation to the flow wake is considered as the most dominant cause to increase mixing in the laminar flow. The larger the vorticity and the longer the rotational region will improve the mixing characteristics.

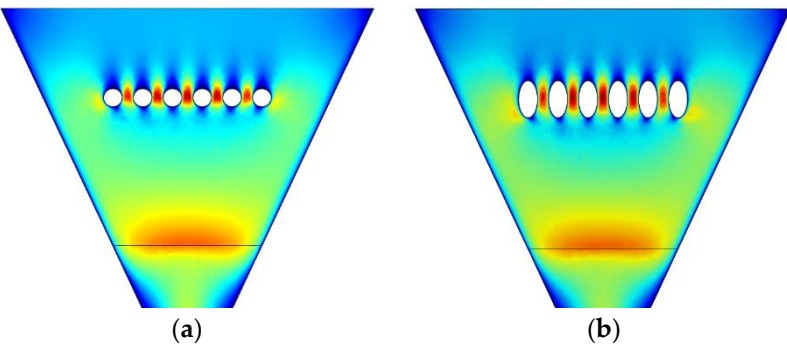

**Figure 13.** Velocity contour at the diffuser: (**a**) Case 2 and (**b**) Case 8.

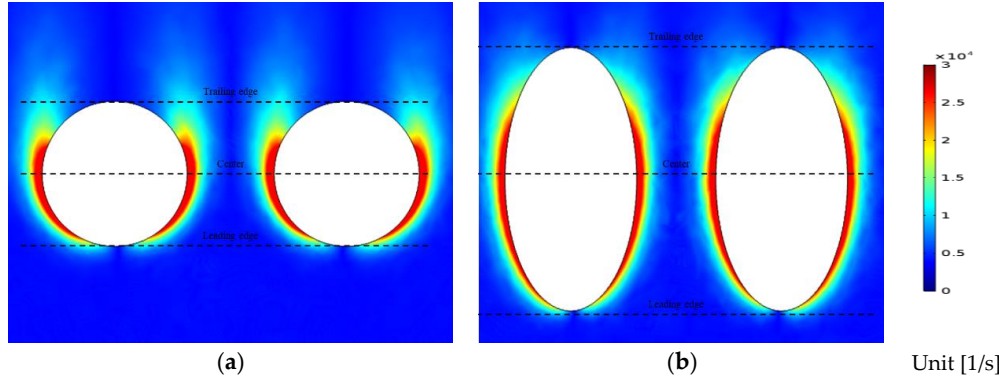

**Figure 14.** Vorticity contour around the obstacles: (**a**) Case 2 and (**b**) Case 8.

For a quantitative comparison, two lumped parameters are proposed as follows:

$$I_p = \frac{\Delta p_8}{\Delta p_2} \tag{25}$$

$$I_\omega = \frac{\int \omega_8 dS}{\int \omega_2 dS} \tag{26}$$

where the subscripts 2 and 8 represent Cases 2 and 8, respectively. Equation (25) indicates the overall ratio of pressure drop where the symbol $\Delta$ indicates the difference of values measured on the discharge of pipe and the outlet of diffuser, and Equation (26) is the ratio of integrated vorticity, or circulation for the estimation of flow rotation where $dS$ stands for the area segment on the surface of all the obstacles. $I_p = 1.03$ means that the pressure drop after elliptic cylinder increases only 3% compared with that of the circular cylinder. However, the integrated vorticity increases as much as 49% since $I_\omega = 1.49$. Consequently, the choice of a 2:1 ellipse shape can be even better than a cylinder because it can increase the overall mixing performance in spite of very minor increase on the adverse effect of pressure loss.

### 4.3. Effect of Crossflow in the Three Dimension

As the height of diffuser is short in the *z*-direction, the only observation at the *xy* plane is not regarded as proper because the flow must be three-dimensional in the narrow channel. Figure 15a,b represents the magnitude of vorticity from Figure 14a,b along the transverse section at the position of leading edge, center, and trailing edge of obstacles, respectively. The three components of vorticity are denoted as $\omega_x$, $\omega_y$, $\omega_z$, marked in Figure 16.

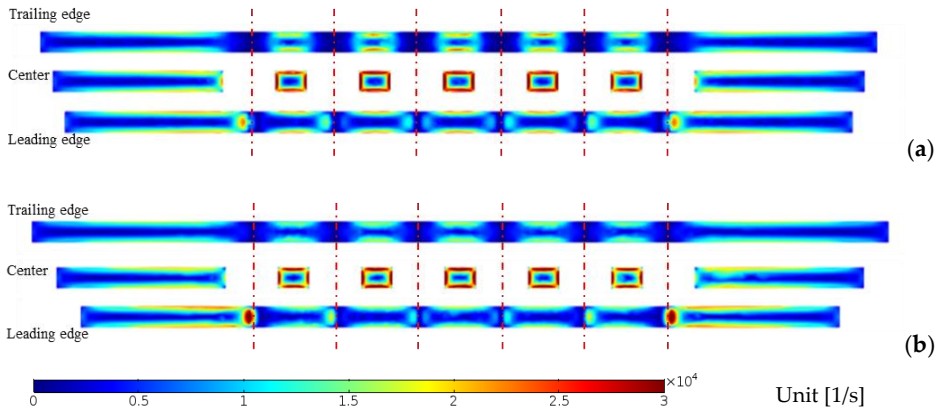

**Figure 15.** Vorticity magnitude for (**a**) Case 2 and (**b**) Case 8.

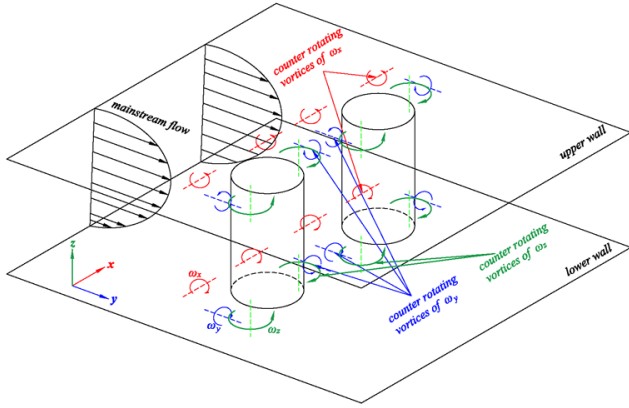

**Figure 16.** Schematic of the three-dimensional vortices around obstacle.

The $\omega_z$ effects obviously on the two-dimensional uniformity mentioned in the previous section, but the narrow gap between upper and lower plates of diffuser results in $\omega_x$, which is almost independent on the existence of obstacles. However, the crossflow or three-dimensional effect can be found in Figure 15a,b where it is denoted as $\omega_y$, consisting of a horse sue figure of vortex filament with $\omega_x$ in Figure 16. The strength of them is compared in Figure 17a–f where $\omega_y$ is only marked significant at the leading edge position, but is diffused at the downstream: see Figure 17b,e. In Figure 17a–f, the magnitudes of $\omega_x$ and $\omega_z$ seems to be similar orders. Overall, the magnitudes are compared as $\omega_x \approx \omega_z > \omega_y$, and, therefore, the vorticity components $\omega_z$ (major) and $\omega_y$ (minor) contributes to the mixing with the installation of obstacles inside the diffuser.

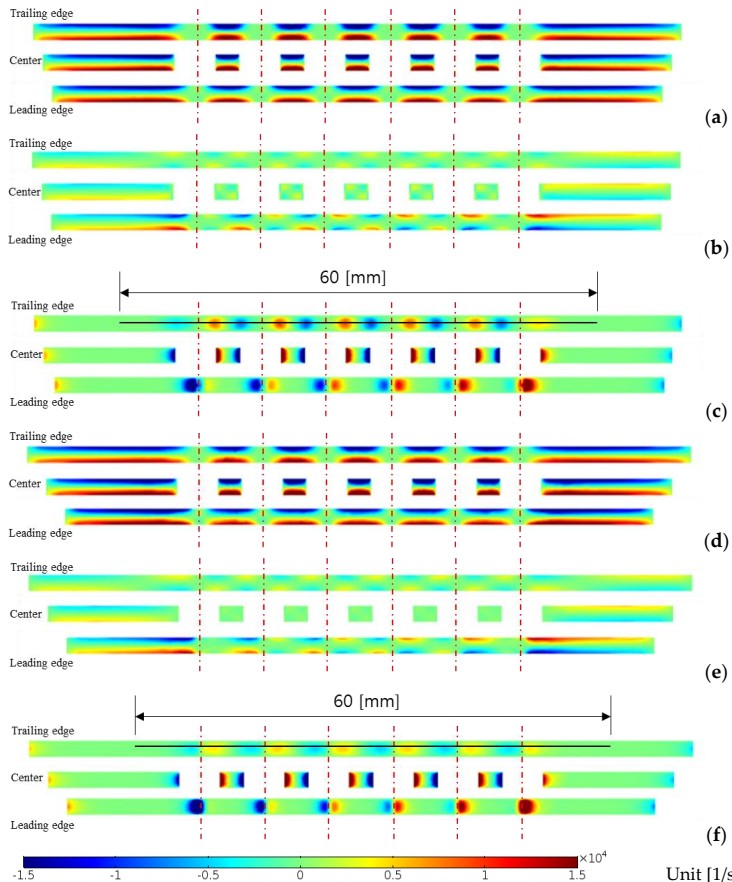

**Figure 17.** Vorticity components of cross flow: (**a**) $\omega_x$, (**b**) $\omega_y$, (**c**) $\omega_z$ in Case 2; (**d**) $\omega_x$, (**e**) $\omega_y$, (**f**) $\omega_z$ in Case 8.

Figure 18 is the comparison of $\omega_z$ at the trailing edge of obstacles for case 2 (circle) and 8 (ellipse). The difference of vorticity components shows a remarkable difference between two cases: the amplitude of $\omega_z$ in the ellipse shape is approximately half of that in the circle shape. Thus, this elucidates the better characteristics of flow uniformity when we used 2:1 elliptic cylinders: see Figure 12.

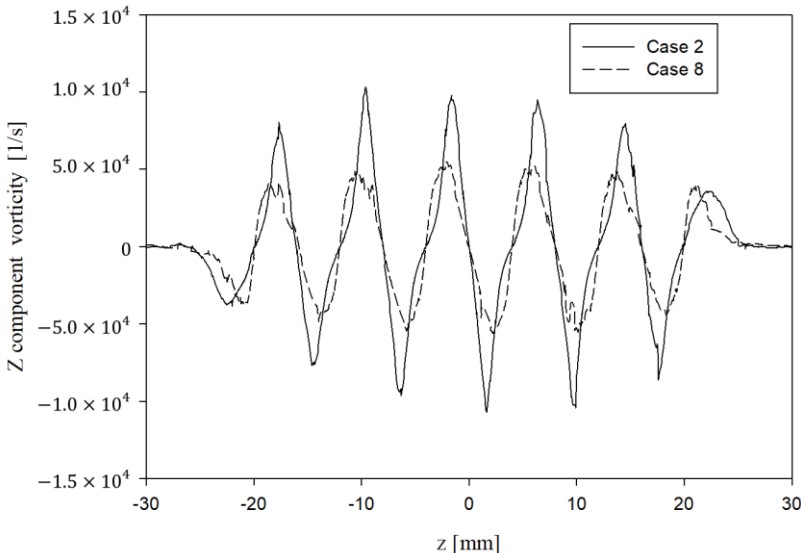

**Figure 18.** Comparison of *z*-component vorticity in Case 2 and Case 8.

## 5. Conclusions

In this research, the flow uniformity and the increase of vorticity are analyzed with computational fluid dynamics applying non-Newtonian fluid model in the range of laminar flow whose Reynolds number is 58.5 based on the inner diameter of feeding pipe.

The remarkable findings on the present study are summarized as: with an inline transverse configuration of resistive obstacles and the control of gap interval, 12 cases are tested for numerical analysis, and finally an optimal case is determined for the uniformity of outlet flow as 60% of gap to diameter ratio and 2:1 ellipse shape. This result is not so different from that of Newtonian fluid [2].

For a proper analysis with RMS values, the effect of boundary layer should be filtered, generated from two side walls in the diffuser. For the purpose of this, the boundary layer of two-dimensional non-Newtonian flow is calculated with a type of Blasius similarity solution, and the boundary layer thickness at the outlet is computed as 30 mm for the baseline case. From the RMS analysis excluding the effect of boundary layer at the side walls, the RMS of velocity deviation at the diffuser outlet indicates 77% decrease in Case 2 of circular cylinders and 92% decrease in Case 8 of 2:1 elliptic cylinders compared with the baseline case. If we just change the cross-sectional shape circle to longitudinal ellipse, the RMS error decreases no less than 15% points. The reason is analyzed that the smoother flow along the surface of body is extended at the wake region due to suppress of vortex shedding originated from the flow separation.

The increases of pressure drop and vorticity are investigated for the best cases of Cases 2 and 8, when the cross sectional shape is changed from circle to ellipse, the pressure drop is increased just in minor 3%, but the vorticity at the body surface increased 49%, which means a remarkable improvement of mixing efficiency with a small effort. That is, the final configuration seems to be very effective for the enhancement of mixing properties for the laminar flow regime, and therefore it has so wide possibility of engineering application on various mechanical parts concerning low Reynolds number fluids.

The analyses of flow vorticities at the cross sections around the obstacles results in overall $\omega_x \approx \omega_z > \omega_y$ for the order of magnitude, which are interpreted as the effect of diffuser walls, obstacles, and crossflow, respectively. The vorticity, $\omega_z$ drives primarily the supply of mixing energy to the main

flow to enhance the mixing of outlet flow. When the z-axis component is compared for Cases 2 and 8 at the line cutting trailing edges, the amplitude was decreased almost half for the 2:1 ellipse case compared with the circle case, which has more uniform velocity distribution at the outlet.

## 6. Patents

A patent related with the present study has been applied to Korea Patent Office (1020180051213) on 3 May 2018, and open to the public (1020190056945) on 27 May 2018, entitled "Nozzle having resistance object for uniform flow in the poly-urethane coating process."

**Author Contributions:** Conceptualization, Y.W.S. and S.-M.C.; methodology, Y.W.S.; software, Y.W.S.; validation, Y.W.S.; formal analysis, Y.W.S.; investigation, Y.W.S.; resources, Y.W.S.; data curation, Y.W.S.; writing—original draft preparation, Y.W.S.; writing—review and editing, S.-M.C.; visualization, Y.W.S.; supervision, S.-M.C.; project administration and research advice, J.H.L.; funding acquisition, S.-M.C. All authors have read and agreed to the published version of the manuscript.

**Funding:** This work was supported by the Human Resources Development Program (Grant No. 20194010201800, 20194010000110) of the Korea Institute of Energy Technology Evaluation and Planning (KETEP) grants funded by the Korea government (Ministry of Trade, Industry and Energy).

**Acknowledgments:** The authors appreciate to KETEP on the heavy pressure for publication as well as its financial support for this research.

**Conflicts of Interest:** The authors declare no conflict of interest.

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
