# Peer review of "Characterization and Control for the Laminar Flow of Liquid Polyurethane System in a Wide Angle Diffuser with Transversely Arrayed Obstacles"

_applsci, doi:10.3390/app10041228_

Round 1

Reviewer 1 Report

The presented paper entitled ‘Flow Uniformity and Mixing Characteristics for the Non-Newtonian Laminar Flow of Polyurethane Pre-polymers in the Diffuser with Installation of Transversely Arrayed Obstacles’ seems to be interesting point of view of urethane pre-polymer processing by injections methods and belongs to theoretical works.

Keywords: High-Viscous Fluid, Flow Uniformity, Polyisocyanurate Board, Mixing Performance, Non-Newtonian Fluid lines 23-24

Instead ‘Polyisocyanurate Board’ should be ‘Poyurethane Board’

Authors wrote “Generally, a hard urethane board is manufactured through the four-step process: mixing, reactions, injection, coagulation, and cutting.” Lines 29-30

Mixing, injection, coagulation and cutting are technological operations not process.

In the introduction part Authors wrote: “In this study, a commercial code for the numerical analysis, COMSOL Multiphysics 5.3a is used for the whole simulations for the injection diffuser flow in the manufacturing process of poly-urethane foams. The urethane pre-polymer has the characteristics of pseudo-plasticity as a  non-Newtonian fluid, so a model of power law is applied for this numerical analysis.” lines 57-60.

The phrase “urethane prepolymer” is incorrectly used. In the case of mixing all components together there is liquid polyurethane system. Urethane prepolymer is oligomer based on polyol and high excess of isocyanate component. The viscosity and rheological behavior of urethane pre-polymers and polyurethanes systems depend on the viscosity of used components to theirs synthesis. In the case of foams, every single component should be analyzed first separately, and next as a mixture.

Author Response

The presented paper entitled ‘Flow Uniformity and Mixing Characteristics for the Non-Newtonian Laminar Flow of Polyurethane Pre-polymers in the Diffuser with Installation of Transversely Arrayed Obstacles’ seems to be interesting point of view of urethane pre-polymer processing by injections methods and belongs to theoretical works. Q1 : Keywords: High-Viscous Fluid, Flow Uniformity, Polyisocyanurate Board, Mixing Performance, Non-Newtonian Fluid lines 23-24, Instead ‘Polyisocyanurate Board’ should be Polyurethane Board’. A1: We agree. Correction has been made. Q2: Authors wrote “Generally, a hard urethane board is manufactured through the four-step process: mixing, reactions, injection, coagulation, and cutting.” Lines 29-30 Mixing, injection, coagulation and cutting are technological operations not process. A2: We agree. Correction has been made by simply deleting the word, ‘process’. Q3: In the introduction part Authors wrote: “In this study, a commercial code for the numerical analysis, COMSOL Multiphysics 5.3a is used for the whole simulations for injection diffuser flow in the manufacturing process of poly-urethane foams. The urethane pre-polymer has the characteristics of pseudo-plasticity as a non-Newtonian fluid, so a model of power law is applied for this numerical analysis” line 57-60. The phrase “urethane prepolymer” is incorrectly used. In the case of mixing all components together there is liquid polyurethane system. Urethane prepolymer is oligomer based on polyol and high excess of isocyanate component. The viscosity and rheological behavior of urethane pre-polymers and polyurethanes system depend on the viscosity of used components to theirs synthesis. In the case of foams, every single component should be analyzed first separately, and next as a mixture. A3: We agreed. We changed the title to “Characterization and Control for the Laminar Flow of Liquid Polyurethane System in a Wide Angle Diffuser with Transversely Arrayed Obstacles’. Also we changed ‘pre-polymers’ to ‘liquid polyurethane system everywhere in the revised manuscript.

Reviewer 2 Report

Research presented by Son et al. presents a new solution for the simulations for the injection diffuser flow in the manufacturing process of polyurethane foams. The article describes an interesting problem that has recently made a significant contribution. The study is systematic, well designed and properly conducted experimentally. The authors present a number of tests. The flow uniformity and the increase of vorticity were analyzed with 294 computational fluid dynamics applying non-Newtonian fluid model in the range of laminar flow 295 whose Reynolds number is 58.5 based in the inner diameter of feeding pipe.
In general, the article has good value in terms of broadening knowledge and expanding research and calculation methods.
However, the document presents minor language shortcomings that need to be corrected. I recommend publishing this article in the Applied Sciences after a small review.

Author Response

Research presented by Son et al. present a new solution for the simulations for the injection diffuser flow in the manufacturing process of polyurethane foams. The article describes an iterating problem that has recently made a significant contribution. The study is systematic, well designed and properly conducted experimentally. The authors present a number of tests. The flow uniformity and the increase of vorticity were analyzed with 294 computational fluid dynamics applying non-Newtonian fluid model in the range of laminar flow 295 whose Reynolds number is 58.5 based in the inner diameter of feeding pipe. In general, the article has good value in terms of broadening knowledge and expanding research and calculations methods. Q1: However, the document presents minor language shortcomings that need to be corrected. I recommend publishing this article in the Applied Sciences after a small review. A1: English has been revised by a native speaker with expertise in the field.

Reviewer 3 Report

Comments are in pdf file

Author Response

Revision of article" Flow Uniformity and Mixing Characteristics for the Non-Newtonian Laminar Flow of Polyurethane Pre-polymers in the Diffuser with Installation of Transversely Arrayed Obstacles"

From the whole text it is difficult to understand what the essence of research is this can only be read partly in verses 241-245, but not in abstract or in conclusions.

The article was prepared not in accordance with the magazine's requirements. You can find the requirements here: https://www.mdoi.com/iournal/aoolsci/instructions#oreoaration.

Affiliation:

Q1: There are no addresses.

A1: We attached the authors’ addresses.

Abstract

Q2: We don't learn anything concrete from the abstract. It was not described what results were achieved, what successes were achieved.

A2: We wrote again the last two sentences:

When the blockage ratio is fixed 0.3 for the pipe of Reynolds number 58.5, eliminating the effect of wall boundary ratio with Blasius velocity profile, the RMS error is reduced 77 to 92% from the baseline case in the case of 60%-diameter gaps for the figure of circles and 2:1 longitudinal ellipse, respectively. The flow is visualized around obstacle components with vorticity as well as flow velocity where the three-dimensional components of vorticity are also elucidated for the evolution of flow wake.

Q3: Line 21: The three-dimensional components of vorticity are also discussed for elucidation of the evolution of f10w wake. I don’t understand the sentence.

A3: Refer to A2 and Fig. 17 where the three components are defined in Fig. 16.

Introduction

Q4: L 37: What means “harness"?

A4: L 43: Typo of “hardness”, and we corrected. Thanks.

Q5: L 48, 56: Unnecessary spaces

A5: L 54, 56, 62: We deleted the spaces between sentence and reference number.

Q6: For purpose of research L 57-60: Why was this model used? Why is better than the previous ones?

A6: Recall that the raw polyurethane system is a non-Newtonian flow. We improved the model from the Newtonian fluid one. The empirical coefficients and powers are referred from the previous references to apply them to our research. So we changed the last two sentences:

As the flow regime of liquid polyurethane system lies in pseudo-plasticity characteristics of a non-Newtonian fluid, a model of power law is applied for this numerical analysis. The flow viscosity is a power function of strain rate with empirical coefficients, which can be edited in the COMSOL source code.

Methods

Q7: In points 2.1-2.4 there are no references.

A7: We attached references: [17], [23], [24] and [25].

Q8: L 121-122 It is not understood whether this is a simulation in a computer program or real tests.

A8: All the data in this paper is obtained from the numerical simulation. The original data is computed for the average flow velocity, and substituted to Eq. (7) to obtain the Darcy friction coefficient.

Q9: 3.2 section-there is no text alignment/justification.

A9: We aligned the text.

Results and discussion

Q10: There are no references.

A10: We attached [18, 22, 24] in L 203.

Conclusions

Q11: It has no right form -not adequate to the magazine's requirements.

A11: We merged them in several paragraphs, changing the form.

References

Q12: Little literature from recent years (2016-2019).

A12: We found a few references related with our topic and recently published in Appl. Sci. journal, and attached them in ref. [26-27].

Q13: Line 384 20. Denham,M.K.,Patrick,M.A. Laminar flow over a downstream-facing step in a two-dimensional flow 385 channel. Trans. Inst. Chem. Engineers 1974, 52, pp.361-367.

A13: Thanks, but this reference is inevitable since it supplies important data for the verification of our model.

Q14: Obsolete position of references. Please replace with a newer one.

A14: The classics are better as academic standards, but accepting the reviewers suggestion, we attached some additional references in ref. [26-27].

Q15: In addition,no standards are given throughout the text. Chapters 4.2 and 4.3 explain the heart of the matter. But this needs to be more specific in the introduction and summary.

A15: We attached some sentences in abstract and introduction, and change the style in conclusion.

Round 2

Reviewer 3 Report

The authors still did not comply with the reviewer's comments.

Author Response

Dear reviewer:

In the preparation of the third version, We revised abstract, introduction, and conclusion again to reflect the findings in Section 4.2 and 4.3.  Especially, the language is revised again very thoroughly.

The standards in each sections are well considered and organized to fit all the paragraphs in conculsion matched to each sections in Chapter 4. The reference of measuring location is written in the text, and the baseline case (with no obstacle) is highlighted in Section 4.1.

We think your kind advice has improved far better our manusctipt, and the authors appreciate it.  We marked all the changed part with red letters. Please check them in the revised manuscript.

We hope this revision to satisfy the high standards in Applied Science.  If you find a new faults in manuscript, please let me know to correct. 

Sincerely,

The corresponding author

Se-Myong(Sam) Chang

Round 3

Reviewer 3 Report

Dear Authors

The authors corrected, supplemented the deficiencies, clarified some unclear issues. The publication can be accepted in its current form.